# Robustness and Flexibility of Neural Function through Dynamical Criticality

**DOI:** 10.3390/e24050591

**Published:** 2022-04-23

**Authors:** Marcelo O. Magnasco

**Affiliations:** Laboratory of Integrative Neuroscience, Rockefeller University, 1230 York Avenue, New York, NY 10065, USA; magnasco@rockefeller.edu

**Keywords:** dynamical criticality, flexibility, dynamic change in function

## Abstract

In theoretical biology, *robustness* refers to the ability of a biological system to function properly even under perturbation of basic parameters (e.g., temperature or pH), which in mathematical models is reflected in not needing to fine-tune basic parameter constants; *flexibility* refers to the ability of a system to switch functions or behaviors easily and effortlessly. While there are extensive explorations of the concept of robustness and what it requires mathematically, understanding flexibility has proven more elusive, as well as also elucidating the apparent opposition between what is required mathematically for models to implement either. In this paper we address a number of arguments in theoretical neuroscience showing that both robustness and flexibility can be attained by systems that poise themselves at the onset of a large number of dynamical bifurcations, or *dynamical criticality*, and how such poising can have a profound influence on integration of information processing and function. Finally, we examine critical map lattices, which are coupled map lattices where the coupling is dynamically critical in the sense of having purely imaginary eigenvalues. We show that these map lattices provide an explicit connection between dynamical criticality in the sense we have used and “edge of chaos” criticality.

## 1. Introduction

Long-term survival requires—by definition—surviving many short terms. This creates a well-studied catch-22 in evolutionary biology: becoming too good at short-term survival, for instance by overspecialization, is detrimental to long-term survival, for which generalist abilities are required in a changing evolutionary landscape. Thus species need to do “well enough” in the short term, but not at the expense of their ability to change strategies when the niche shifts. In physiology, these two conflicting temporal demands are loosely identified with “*robustness*”, the ability of a physiological system to perform the exact same task correctly under varying and uncontrolled conditions, and “*flexibility*”, the ability to change the task as conditions change. A well-established example is the heartbeat, which needs to be robust in the scale of minutes (identical and well-spaced heartbeats no matter what the heart rate) but remain flexible in the scale of a day (rapidly and sensitively adapt heart rate to physiological demand).

While a number of studies have explored the theoretical concept of robustness as it pertains to various areas of biology, most prominently in molecular cell biology [1,2], the theoretical underpinnings of biological flexibility are still obscure [3]. It is known how to show that a mathematical model of a biological system is robust, as there are well-defined prescriptions to show the solutions to be insensitive to small perturbations in the defining dynamics. However, it is not so clear at present what mathematical procedure we should follow to determine if a model is *flexible* or not, a difficulty due in no small part to the apparent loggerheads between these two notions. Here, we suggest that a family of strategies (“dynamical criticality”) provides a foundation for general recipes for flexible yet robust functions.

One of the most striking forms of flexibility in neural function is *integration*. This phenomenon occurs at various scales, from the input-dependent changes in the range of intracortical functional connectivity, all the way up to entire brain areas working together as needed, forming transient associations. Modeling these dynamical changes in the size and range of interactions is a key part of this area of research.

In this Introduction, we first survey the historical development of these ideas within the neuroscience community, briefly describe some nuances between the different senses in which criticality is used in various contexts, define carefully the notions of robustness and flexibility, and review the mathematical notion of “general position” on which structural stability theory of dynamical systems is predicated.


**Historical development**


One can always find in the classics prefigurations of things to come, because scientists vaguely intuit ideas well before the technical developments exist to discuss them rigorously. The year 1948 was rife with prefigurations of the ideas that eventually would be dynamical criticality, such as Ashby’s construction of the *Homeostat* [4], built on the ideas of *Principles of the Self-Organizing Dynamical System* [5], and Hopf’s treatment of the Landau–Hopf theory of turbulence [6]; but the most explicit one is Thomas Gold’s absolutely prescient 1948 paper *The physical basis of the action of the cochlea* [7] where he notes that the degree of mechanical resonance that can be measured in the cochlea is incompatible with the heavy viscous damping expected of a fluid system with narrow passageways, reacting purely passively to mechanical forcing. He posited the existence of an active mechanism which would provide enough “negative viscosity” to overcome the physical viscosity of the fluid mechanics. He further predicted that if the active mechanism overcompensated, it could result in a spontaneous feedback oscillation and emission of sound *from* the cochlea.

However, his theory rested on a single experimental result: his own. He and Pumphrey had measured the degree of resonance in question, through extremely ingenious yet entirely indirect means, in the article immediately preceding reference [7] in that issue of the journal [8]. Actual mechanical measurements in physiologically intact cochleae would not be technically feasible for many decades. Thus, his predictions were not taken seriously, especially those about sound coming out of the cochlea, and all attention in the cochlear dynamics field went to von Békésy’s measurements of traveling waves in the cochleae of cadavers, which, by virtue of being dead, cannot show an active process powered by metabolic energy; thus, none of von Békésy’s work could therefore prove or disprove Gold’s conjecture. It was not until the late 1970s that Kemp discovered the *cochlear active process* [9]—briefly thereafter, spontaneous otoacoustic emissions—the nearly-universal phenomenon where, in a very quiet environment, sound can be measured coming out of most healthy cochleae. Even then, However, Gold’s ideas remained at the level of purely qualitative discussions.

A total of 50 years after Gold, in 1998, while constructing a biophysically plausible model of the dynamics of *hair cells* (the sensorimotor cells in the cochlea responsible for the active process) we noted [10] that choosing physiologically plausible parameters seemed to poise our model close to a Hopf bifurcation. Not only that, but upon changing the length and number of stereocilia in the model, in accord with extant electron-microscope measurements, the *frequency* of the Hopf bifurcation in our model tracked the physiologically measured frequency of cells of those characteristics. We further noted that if the model was tuned *exactly* to the Hopf bifurcation, then when subject to small forcings (i.e., faint sounds), the model’s response was narrowly tuned in frequency and showed a large gain, both of which characteristics vanished for larger forcings, which were previously known yet unlinked experimentally observed features of the hearing system.

In modern language we now recognize that what Gold had been describing was a Hopf bifurcation, together with a hypothetical mechanism that would tune the parameters of the system to be close to such a bifurcation. Up to our work, the active process’ action was only described in qualitative terms, “injecting energy into the system”. By following Gold’s conjecture *literally* rather than figuratively, poising the system *exactly* at the bifurcation, precise theoretical predictions could be made. Briefly after our work, Camalet et al. [11] conjectured mechanisms to tune a system to being close to a Hopf bifurcation, which they termed “self-tuned criticality” to distinguish it from the different notion of “self-organized criticality” from sandpile models. Later we showed [12] that a reduction to a normal form for the forced Hopf bifurcation displayed four characteristics that vanish away from the critical point: high amplification, high spectral tuning, a compressive nonlinearity of the response following a 1/3 power law, and, if mistuned, the ability to go into a limit cycle and generate oscillations. The theoretical advance here is that the Hopf bifurcation scenario links all four of these independently known characteristics as part of one single scenario. Moreover, following Gold, we need to posit the existence of a feedback loop to keep the system tuned to the bifurcation [13,14].

To this day, the Hopf bifurcation scenario remains the best extant theoretical description of the sensitivity and acuity of our hearing [15]. No experimental manipulation of the organ of Corti has been able to ablate one of the four characteristics without ablating all four. The 1/3 power law has been experimentally measured in a number of systems such as sacculus cells [16]. The specific nonlinearities giving rise to the 1/3 law have also been shown to account for the special scaling of $\left(2f_{2}-f_{1}\right)$ combination tones [12,17], two-tone interference [17,18], and various other phenomena which, in other models, require separate scenarios to explain.

Meanwhile, on the (then) other side of neuroscience, *motor control*, David Tank’s group had been making a number of measurements in the oculomotor reflex of the goldfish that appeared to defy any classical theoretical explanation. Sebastian Seung posited a remarkable model [19]: a neural system that could poise itself so that its attracting set, rather than being a fixed point, was a *line.* Such line attractors, by virtue of having one direction (along the line) in which their stability is indifferent, are able to perform exact integration of inputs when mapped as forces *along* the line. Line attractors can be generated when one of the eigendirections around a stable fixed point loses stability and becomes marginal, its eigenvalue becomes zero, and its associated stable manifold becomes a center manifold. Evidently, to achieve a neutral direction a parameter must be tuned and therefore a feedback control mechanism must exist. In a striking series of experimental demonstrations, Tank’s group showed that pharmacological manipulations of the goldfish circuit interfered with this poising and rendered the central rest position of the goldfish’s eyeball (the fixed point) either stable or unstable [20,21,22,23], so that in order to foveate an off-center target the system had to repeatedly perform saccades returning to the target position.

Roughly in the same timeframe, Elie Bienenstock was proposing the notion of “regulated criticality” [24], in which he used a form of Hebbian covariance plasticity for the feedback equations keeping the system close to a critical point. He analyzed a low-dimensional E-I system whose synaptic parameters were then “regulated” through Hebbian covariance.

A model of motor “gestures” [25] analyzes vocal apparatus of songbirds, pointing out that when the vocal system is poised close to a dynamical bifurcation (in that case a Hopf transition), very simple parameter trajectories in parameter space (“gestures”) can generate extremely complex output waveforms. See Figure 1. This line of research anticipates the “rotational dynamics” found in, for example, motor reach actions [26] and song premotor cortical neurons in songbirds [27].

Our historical tour ends with a model poised closed to a critical transition so that cyclical motion in parameter space enables it to reconfigure its dynamics to perform distinct tasks [28]. The model is based on an experiment on sensory discrimination in which a primate carries out a task in which it has to discriminate between two different categories of stimuli, but instead of immediately conveying its judgement, it has to remember this choice for a period of time during which the sensory stimulus is removed, then press a button to convey its answer, and finally forget this past choice in order to start anew another trial. In the model, nearly parallel nullclines are used so that the system can, with small shift of parameters, move from a potential with a nearly-flat bottom (integration, as in a line attractor), bifurcate to a double-well potential (threshold the previous state and then keep the memory of it), and then bifurcate again to a single-well potential (resetting the state of the system to a neutral memory before a new task). See Figure 1. The similarities to the cyclical measurements in Szilard’s famous heat engine, the classical model that established the *kT ln2* thermal equivalence for resetting a bit information, are striking [29,30].

In all these models, we observe a common feature: something *very concrete* is gained by the neural system by living right at a dynamical bifurcation. It can be sensitivity and selectivity (auditory), the ability to integrate over timescales far longer than the biophysical constants of the system (line attractors), the ability to generate and coordinate complex outputs from simpler control “gestures”, or the ability to cycle a system through an integration/decision/memory cycle.

Another feature is that they are all low-dimensional systems. However, the original system under consideration, the brain, is expected to be host to many such critical systems *at the same time.* For example, in the cochlear case, every single frequency band is expected to self-tune to a Hopf bifurcation, so we would have a system with many marginal modes at the same time. In this paper we will refer to “dynamical criticality” as a system in which this is possible, one in the number of critical modes (the dimensionality of the center manifolds) is large and increases with system size. It is not *a priori* evident that this is possible because high-dimensional center manifolds are fragile [31].


**Relationship to other forms of criticality**


There are a large number of interconnected notions of “criticality” that have been used to describe neural systems as well as biological systems in general. Here, we briefly review these other ideas.

Dynamical criticality in the sense of **“edge of chaos”** was a notion advanced by Packard [32] partly inspired by models of Kauffman [33,34], and then further developed by Langton [35,36], Kauffman [37], Mitchell and Crutchfield [38], and others. Other articles in this Special Issue elaborate on the rich evolution of this concept in depth. The concept relates to a system that has an order–disorder transition or order–chaos transition, and the related findings that many systems at this transition are capable of information-processing tasks not available away from this boundary, specifically in the sense of Turing universal cellular automata such as Rule110. The central differences with the concept as used in the current paper are two: first, many of our systems do not have a transition to disorder but a bifurcation to a different dynamical state, and second, we want to consider systems having a large dimensionality of the transition space. In this paper we will show one explicit connection between models in either class, by exploring critically-coupled map lattices.

**Self-organized criticality** refers to a state in which the system spontaneously, and without external tuning of parameters, behaves similar to a statistical mechanics system at a second order phase transition such as the ferromagnetic critical point. Its most classical embodiment is the Bak–Tang–Wiesenfeld sandpile [39]. Several researchers have advanced the notion that the brain is a self-organized critical system in this sense [40] and experimental evidence for “**neuronal avalanches**” has been found in many systems (see [41] for a review). It has been pointed out that existence of a power-law of avalanches does not imply necessarily that the system is critical [42] but further tests on the interconnectedness of various power-laws do lead to methods to measure proximity to criticality in this sense [43,44] Furthermore, experimental evidence for the return of a system after criticality has been perturbed [45,46,47,48] demonstrate the existence of underlying homeostatic mechanisms involved in the maintenance of criticality.

While there are a number of underlying theoretical connections between SOC and dynamical criticality, as shown explicitly in [49], there are also a number of differences. A consequence of SOC is divergent spatial scales, such as an infinitely-long ranged susceptibility. The concept discussed in this paper of self-tuned dynamical criticality refers to a system spontaneously poising itself close to multidimensional bifurcation points. Some systems, such as certain shell models of turbulence, appear to display both kinds of criticality, perhaps for the same underlying reason. However, in the viewpoint we articulate, the nonstationarity of the system is essential; the idea is not to be exactly at a bifurcation point, but to be close enough to one to be able to rapidly change behavior just by moving across the critical value.

**Poising at Statistical Criticality:** Subtly different from the above case, it was observed that, while fitting maximal-entropy models [50] to binned multidimensional spike trains, the fitted Hamiltonians were usually far closer to a statistical critical point than when disturbed or surrogate data was used for the fit. This led to a number of investigations [51,52,53]. Theoretically, proximity to a phase transition allows for increased information transmission capability [50,52,53,54,55].

**Rotational dynamics:** Work on motor reach gestures [26] showed that neural activity during an arm-reaching task collapsed on a low-dimensional manifold within which the activity left a quiescent point, rotated around it 180 degrees and re-entered in the opposite side. Different reaches led to different hyperplanes on which this happened, but it was always a rotational motion. Another relevant work is [27], in which these motions are more complex than just half-circles but are nonetheless rotational. Obviously, such rotational dynamics would appear as purely imaginary eigenvalues to an autoregressive analysis. This is one of many cases where we observe the effective dimensionality of brain activity dynamics as being relatively low for any single individual act or task, but with a large variety of different low-dimensional manifolds available.

**Balanced networks:** A different take on excitation–inhibition homeostasis has been taken in a series of studies of “balanced networks”, in which the observation is that, persistently, the *total* amount of excitatory *input* and the total amount inhibitory input are largely balanced together [56,57]. This is a very different criteria to the purely dynamical “imaginary eigenvalue” of the Hessian matrix, as the “total input” can only relate networkwise as products of synaptic strengths times presynaptic activity levels. Balanced networks have been the subject of pretty intense theoretical analysis [58] for their efficiency.


**Robustness versus flexibility**


Consider the heartbeat, once thought the paradigm of regularity. Today the heartbeat is one of the best-documented cases of *1/f* fluctuations in biology, a paradigm of wide variability [59,60,61]. The heartbeat needs to be stable and **robust**, every day of our lives—the heartbeat cannot depend sensitively on fragile combinations of parameters, for we need a heartbeat *always*, regardless of variability in our environment and internal medium. Thirsty, feverish, cool, running, resting, drunk, sleeping, swimming, rushing: we always need a heartbeat, so it needs to be robust for us to live moment to moment. Robustness, in this sense, has been the subject of extensive discussions in the biological literature.

However, if our heart were to beat at the same frequency in any of the above conditions, we would soon be dead. In fact, the heart’s ability to rapidly and sensitively adapt the interbeat interval to behavioral variation in demand for blood supply is a leading indicator of cardiac health. There are a lot of things that happen in one lifetime, so to last through a lifetime of events, we need a flexible hearbeat.

Flexibility and robustness are thus apparently at odds. Consider building a theoretical model of a flock of birds or a school of fish. Making the model “robust” means the agents will stably be attracted to others and easily flock together, but this means that the flock will be governed by majority rules. Making the model flexible means making a flock that is easily steerable by any of its members, a flock that can sensitively change directions when only a handful of individuals may be alarmed by seeing a predator. The solution is that flocks may be both robust as well as flexible when they are, paradoxically, on the borderline of dissolution. Recent work analyzing high-throughput data from the flight of sterling flocks shows that indeed flocks self-poise at critical points as a means to achieve both robustness as well as flexibility [62,63]. 

It has long been implicitly assumed that the biological notion of “robustness” corresponds to the mathematical notion of structural stability; a dynamical system is said to be structurally stable if and only if the qualitative nature of the solutions is unaffected by small perturbations to the defining law (typically the *C*^1^ topology). The quintessential example of a structurally stable system is the stable fixed point, but the very features that make it robust also make it not be flexible. For example, stable fixed points are excellent for remembering things, so they are used in models of memory such as Hopfield networks. However, because they are robust they are hard to change, so it takes many exemplars to train a Hopfield net.

Many demonstrations of structural stability depend upon the dynamical object being described as the intersection between various surfaces, for example, the intersection between nullclines. The system is then structurally stable if this intersection is “generic”. Genericity in topology is one of many mathematical concepts falling under the general umbrella of “general position”.


**General position, structural stability, and robustness.**


Many concepts in mathematics are predicated upon the study of objects in what mathematicians call “general position”, meaning the typical situation that would arise if the objects were randomly chosen or randomly jittered. For instance, three points on a plane *typically* do not lie on the same line, so for them “general position” means “not collinear”. From this point of view, the probability of finding such a “non-generic” configuration, if the three points are randomly and independently drawn, is zero. Many of the most powerful theorems in differential topology only work for *transversal intersections*, namely surfaces that cross each other at a non-zero angle. Two 2D spherical surfaces in 3D typically intersect either on a circle, or not at all; when they intersect on a circle each of the spheres is transverse to the other at the intersection, and a small enough motion of the spheres cannot destroy the circle. If they do not intersect at all, a small enough motion cannot create an intersection. It is only when they barely touch in a point, that the system is critically sensitive to small motions. Transversal intersections all look the same, while the nongeneric, *tangential intersections*, can end up being rather baroque. Because general position leads to powerful tools, and nongeneric systems can become very complex, it is natural to want to assume general position as much more powerful theorems can be proved. So, much of dynamical systems theory assumes it is unlikely to find, in natural settings, dynamical systems which are not in “general position”—for instance, one that has tangential nullclines.

However, in a different setting this may not be at all the case, as illustrated in Figure 2. If three points move randomly around on the plane, one of them *shall,* eventually, cross the line spanned by the other two. If this is the setting we are in, at some *rare* moments in time, it is a *certainty* that general position will be violated. We transferred the rarity of collinearity to a small collection of moments when it happens, but the improbability became a certainty that it will *eventually* happen. A classical example is a dripping faucet: the body of water changes topology from a single bulge connected to the faucet to two disconnected pieces, the bulge and a falling drop, and this change is exceedingly brief, but it still happens regularly. At the very instant of the disconnection, the surface of the faucet-attached bulge and the surface of the drop intersect, but this intersection cannot be transverse. In some cases an infinite cascade of complexity ensues [64,65], whereby the drop is separated from the bulge by a neck of fluid, and the neck itself generates a thinner neck and another thinner neck.

To make matters more interesting beyond brief moments, living beings *regale* us with situations in which a feedback loop *stabilizes* the system at such a non-generic configuration. To belabor the three-point analogy, consider two kids playing hide and seek, and one hides behind a tree; in order to remain hidden the child hiding tries to keep the seeker, the tree, and themselves collinear, with the tree in the middle; this is an active feedback loop stabilizing an otherwise nongeneric configuration. The codimension of the configuration is the number of parameters that need be tampered with by feedback to achieve it (in the sense of the law of requisite variety). This number could become large even for simple cases: a shooter taking aim at a target seeks to make collinear in 3D space (a) their pupil, (b) the target, (c) the front, and (d) the back of the aim’s marks; this system has a codimension of four. When such an active feedback loop is active, lack of general position is no longer a rare event happening at isolated moments in time, but may become the persistent norm; the flip side being that now a feedback loop keeps the system *close to* the nongeneric configuration but *not exactly at it*, and may hover around. I already showed in the Historical Development section a parade of examples of non-generic behavior in the nervous system; the evident conclusion is that somehow, nongenericity is *useful* for living beings. However, this is, mathematically speaking, a wild and lawless territory where few theorems assist us, particularly as the codimension of the system becomes large.

In the last few years, a number of theoretical models and experimental observations, by our group as well as others, have started to paint a coherent picture in which systems whose dynamics is poised on the edge of a sudden change (a “bifurcation”, in jargon) are implicated in the ability of our brains to perform certain tasks.

## 2. Materials and Methods

In this section I will first poise the question of what methodology needs to be used in proving whether a system is “dynamically critical”. Then I describe methodology to extract dynamical eigenvalues from time series. Finally, I examine mathematical models of self-poising leading to dynamical criticality as they are the basis of subsequent studies, and models of cortical integration of information mediated by critical systems.


**Is it a tautological notion?**


There have been recent discussions (mostly in social media) as to whether the notion of being tuned at a critical point is tautological, based on its presence being detected exclusively through model fitting of data. The critics have pointed out that, if the model being fitted to the data has “boring” regimes that are structurally incapable of fitting the data, then if a regression produces parameters that lie at the boundary between these regimes, it is not a proof the “system lives at the boundary”; it has only proved that the only place this particular model can account for the data is at the boundary, but this is a limitation of the model not the original system. In other cases, the observation has been that simple data pre-processing steps such as filtering may further facilitate this process, particularly when looking at the data in a “dimensional reduction” framework.

To be specific, consider trying to fit a generative model to data. In a generative model one does *not* model the outcomes, but instead models the dynamics or process that *gives rise* to the outcomes [66]. For instance, given a set of sequences, one can use a spin Hamiltonian as a model, and then the probabilities of sequences are generated by the Boltzmann distribution; the data can be fit through maximum likelihood. Alternatively, one may have a continuous time-series and fit locally a linear dynamical law as an AR(1) process [67,68,69]. In the case the model is a spin interaction Hamiltonian, then any amount of nontrivial structure in the sequences will preclude the fit from using model parameters in a high temperature “gas” phase where correlations fall exponentially; nor in the ferromagnetic phase where sequences have little variability. If the Hamiltonian has a phase transition, that is where freedom to fit the observations will live. Or in the case of fitting linear dynamics to a time-series, linear dynamics can decay, grow, or oscillate. If the window of the fit is long and the time series is approximately stationary, then the fit will not produce exponentially decaying or growing functions, since these are nonzero only at the beginning or the end of the time series; therefore, the only available freedom is to fit oscillations. Monsieur Fourier assured us a long time ago we can always do so, and so coming up with a bunch of oscillatory modes would not be a surprise but an almost certainty.

*This does not mean that it is impossible a priori to prove that a system is near criticality.* This simply means that one has to be careful methodologically. Elsayed and Cunningham give a beautiful treatment of this topic in [70] from the viewpoint of statistical analysis, and argue for a number of tests to be done *a priori* to be certain the interesting “population structure” being observed is not artifactual. Obtaining the model parameters from the fit is only one part: examining the residuals, fitting gently reshuffled surrogate data, should all be part of the standard methodology. In addition, experimental design, and in particular perturbation experiments, should be a part of this methodology from the start, as critical systems respond to perturbations in characteristic fashions; for example, Tank and collaborators observed directly a switch to either exponential growth or exponential decay when performing pharmacological manipulations. In [67,71,72], the point is not that the eigenvalues of a fitted dynamics live close to the critical line in conscious patients, but that induction of anesthesia drives these eigenvalues to a more stable regime: it is therefore directly probatory of the idea that nearness to the critical line is not entirely an artifact of the fit. Similarly, in [73], proximity to the critical line is only observed in healthy patients while pathologies show stabilization of cortical dynamics.

Turning the tables around, one may argue there is no such thing as a model that gives interesting dynamics for all parameters, therefore the criticism is itself a bit of a tautology.


**Self-poising in anti-Hebbian networks:**


In our first paper on dynamical criticality in neural networks [49], we outlined a model that serves more as an “existence proof” than as an actual proposal: a set of neurons whose dynamics is simply linear in the activities, but whose synaptic matrix evolves through “anti-Hebbian” interactions [74,75,76,77,78,79,80,81]:(1)$$\dot{x}=Mx\;\dot{M}=\alpha \left(I-{\mathrm{xx}}^{T}\right)$$

The evolution of this system is surprisingly complex. As shown in the Figure 3, the real part of the eigenvalues of *M* relaxes towards 0, so that all eigenvalues gravitate towards the instability boundary (the imaginary axis) and forever thereafter flutter around it.

This model showed that dynamics that stabilize the system around many simultaneous Hopf bifurcations can indeed exist, pushing the cloud of eigenvalues and then flattening it against the imaginary axis.


**Integration:**


A characteristic of Hopf bifurcations that we first derived in [12] in the context of hearing and modeling of hair cells, is that a system poised at the Hopf bifurcation can amplify very faint signals yet attenuate loud signals, a feature called *compressive nonlinearity*. This is achieved by displaying a timescale of integration that is dependent on the input: the system integrates over many cycles if the input is small, so as to amplify, yet relaxes to its final state rather rapidly if the input is large. In [82], a study of signal propagation in primary visual cortex determined that signals propagate outwards from an epicenter and attenuate very slowly if the stimulus has low contrast, while stimuli of high contrast led to rapid exponential decay of signals, showing that the functional connectivity of V1 is directly modulated by signal strength. We created a model, poised at the critical point, and demonstrated that it possesses precisely these features [83].

**Critical Mode Analysis:** Inspired by the model above, we undertook to see whether adopting this model as a point of view from which to analyze data would be fruitful. We proposed in [68] to analyze electrocorticography (ECoG) data in human subjects by proposing to reconstruct the underlying law of motion of the activity to the lowest possible order, namely linear. Thus, if the ECoG data is represented as **x**(*t*) where **x** is a vector containing the voltages observed in the array, and *t* is time, we would want to posit that
$$\dot{x}\left(t\right)=Mx\left(t\right)+\eta \left(t\right)$$
and we would try to regress the matrix *M* by minimizing the residuals η(t) in some norm over a snippet of time brief enough to have good resolution but long enough to reliably populate the statistics. Finally, the eigenvalues of *M* are observed in the complex plane; their imaginary part is an oscillation frequency, while their real part is an exponential attenuation (negative real part) or an exponential growth factor (positive real part). To our surprise, the eigenvalues came remarkably close to the instability line, and every form of surrogation we tried, from strong to subtle, disrupted this closeness.

Technically, our analysis is naturally carried out in discrete time, as the time series is sampled by the ADC at some regular sampling rate, and so what we fit is called in statistics an AR(1) process; the main difference between our use and the classical one being we resolve it in time by fitting in small windowed snippets. A study of brain activity during a finger-tapping task first presented this method in [68].

**Critically coupled map lattices**. Consider the following equation
$${\dot{x}}_{i}=f\left(x_{i},y_{i}\right)+\underset{j}{\sum }M_{ij}x_{j}$$$$\dot{y}=g\left(x_{i},y_{i}\right)$$
where *M* is a coupling (“synaptic”) matrix poised at criticality. Such equations were shown [84] to be able to display extremely complex spatiotemporal behavior reminiscent of complex cellular automata, and explicitly realize a connection between the notion of criticality we have hitherto put forward and the “edge of chaos” criticality [35,37,38,85]. Such equations have also been recently studied from the point of view of recurrent neural network theory [86], since the purely imaginary eigenvalues solve the issue of backpropagation gradients blowing up in training of the recurrent neural network.

In order to simplify the analysis and the computational cost of simulations, it was proposed in [84] to construct coupled-map lattices by using the following conceit. Imagine that the intra-unit dynamics are allowed to operate without coupling for a period, and then the coupling is allowed to operate, without internal dynamics, for a period. The operation of the dynamics in parallel in each unit can be described by a stroboscopic map to intermediate variables
$$\xi_{i}^{t+1}=f\left(x_{i}^{t}\right)$$
and then the operation of the coupling without internal dynamics will show up as a convolution with a matrix *G*
$$x_{i}^{t+1}=\underset{j}{\sum }G_{ij}\xi_{j}^{t+1}$$
where *G* is the matrix exponential of the coupling matrix *M* times the integration period τ:$$G=e^{\tau M}$$
from where we can explicitly now see that if *M* has purely imaginary spectrum, and in particular, if *M* is antisymmetric, then *G* is unitary and preserves phase space volume. The rich computational properties of this family of dynamics are then in no small part a result of the couplings not contracting or dilating phase space volume: all phase space loss is in the hands of the dynamics of individual units. Furthermore, if the original couplings *M* are local, then the range of *G* grows with τ but the values asymptote to zero at long distances. If *M* has a convolutional structure, then *G* is also a convolution.

It is instructive to look at the actual couplings to have a clear idea. For example, if one considers a 1D lattice in which nearest-neighbor coupling follows a “checkerboard” rule of the form $M_{i,i\pm 1}={\left(-1\right)}^{i}$, the exponential of the matrix is shown in Figure 4:

The behavior of a critically coupled map lattice in 1D with map $f\left(x\right)=\alpha xe^{-x^{2}/2}$ and $M_{i,i\pm 1}={\left(-1\right)}^{i}$ was described in [84]. Here we describe further work on 2D lattices.

There are obviously many ways to create antisymmetric couplings, even translationally invariant ones. An obvious one is to use the checkerboard coupling, where every odd site is excitatory and every even site inhibitory and all couplings have the same magnitude and opposing signs. This coupling results in a strongly anisotropic dynamics with gliders moving along diagonal directions; see Appendix A. Ways to create a less anisotropic lattice would clearly be desirable.

An elegant way is to use a bipartite graph; the desired lattice is made in two copies, and the top (excitatory) lattice only couples to the bottom (inhibitory) lattice and vice-versa. We will present below results pertaining to circularly symmetric Gaussian couplings with kernel $e^{-\frac{\Delta x^{2}+\Delta y^{2}}{2}}$.

## 3. Results

**Anesthesia:** The only commonality to anesthetic drugs is that they produce a **reversible coma**. Anesthetics have no common molecular structure, binding targets or brain structures; nor do they produce similar effects on neural population activity on EEG or LFP. It is not *a priori* evident that once the brain is perturbed to an unnatural state it should ever return on its own to a normal state, and in fact many comas are not reversible for no apparent reason. However, patients routinely recover from extremely deep anesthesia. Recovery from the anesthetized state is not a purely pharmacokinetics phenomenon: it displays steep hysteresis and history-dependence, so it is impossible at intermediate concentrations to predict whether a subject is awake or anesthetized from drug concentrations alone. We used the above described critical mode analysis to carry out studies in humans [72] and monkeys [67,71] which have demonstrated that across anesthetic agents and across species and subjects, a constant in loss of consciousness is *the stabilization of cortical dynamics*. See Figure 5.

These studies, amongst others, suggests recovery from deep anesthetic coma may be intrinsically activity-dependent. Data suggesting the rate-limiting steps in recovery are not pharmacokinetic but related to transitions in ongoing brain activity was shown in [87].


**Critically Coupled Map Lattices**


The dynamics of a critically coupled map lattice is given in two steps: in step 1, variables x at every site i are transformed into intermediate variables $\bar{x}$ by application of a 1D map f(x), “in parallel”. In the second step, the variables $\bar{x}$ are coupled together using a matrix U to generate the original variables x at the next timestep:$${\bar{x}}_{i}^{t}=f\left(x_{i}^{t}\right)$$
$$x_{i}^{t+1}=\underset{j}{\sum }U_{ij}\;{\bar{x}}_{j}^{t}=\underset{j}{\sum }U_{ij}f\left(x_{i}^{t}\right)$$
where
$$U=e^{\lambda M}$$

As stated above, the behavior of a critically coupled map lattice in 1D with map $f\left(x\right)=cxe^{-x^{2}/2}$ and $M_{i,i\pm 1}={\left(-1\right)}^{i}$ was described in [84]. In this work it was shown the use of dynamically critical couplings (i.e., a dense number of modes set precisely to be at the onset of a Hopf bifurcation), when used stroboscopically in conjunction with a simple map capable of classic transitions to chaos by period doubling, give rise to a coupled map lattice showing enormous variety of complex behaviors, some of them strongly reminiscent of the universal cellular automata at the forefront of “edge of chaos” arguments. See Figure 6. The important difference with cellular automata is that these are in fact continuous dynamical systems having parameters that can be continuously varied in order to generate a huge variety of behaviors. Analysis of the set of parameter values displaying complex spatiotemporal behaviors (neither orderly nor chaotic, but having complex long time and long-distance dynamics) is a solid region in parameter space; i.e., parameters need not be tuned to a rather specific combination in order for these complex dynamics to be robustly displayed, explicitly making the connection between robustness and flexibility, the central themes of this article.

To analyze this complexity one way is to gather spatiotemporal spectra of the model’s behavior—Fourier transforming in both space and time and averaging over many spectra. Taking such power spectra as probability distributions, their entropy characterizes the complexity of the responses. Synchronized chaotic behavior has white spectra along time but no frequency components in space; frozen disorder has spectra along space but not time. Behavior that has complex long-range and long-time structure has a large fraction of the power at low, but non-zero, spatiotemporal frequencies.

As described in M&M, the simplest way to generalize the 1D model by using a 2D checkerboard lattice generates a system with strong anisotropy along the diagonals and a complex set of gliders, as displayed in Appendix A. We then moved to a bipartite lattice system using a circularly symmetric Gaussian coupling between sites. One recurring asymptotic behavior in this model consists of a background of frozen dislocated checkerboards as seen in Figure 7 (left). In a portion of the phase space the dynamics halts at these frustrated ground states, suggesting the model has a large number of different stable fixed points; but close to this region small vortical structures (showing up as darker shades in the figure) move around similar to gliders in cellular automata; they re-write the ground state in the wake of their passage. (See Appendix A). An overall phase diagram is shown in Figure 8. 

## 4. Discussion

For a long time, basic logic as well as careful measurements have suggested that the amounts of excitatory and inhibitory activity in the brain must be carefully balanced to avoid entering either uncontrolled oscillations or, alternatively, total loss of activity. This is one of potentially many homeostatic regulations the nervous system operates under. As we have described, recently, a more specific idea has taken shape: data from sensory and motor cortices as well as theoretical models have suggested that mechanisms poise neuronal activity at the verge of dynamical instabilities, and are involved in organizing the chatter of individual neurons into coherent large-scale assemblies. This is a more specific form of excitatory/inhibitory balance, where not only the totals are balanced at the global level, but they are balanced in detail in individual patches. Such mechanisms create, virtually by definition, a state of ongoing activity just to be able to regulate the system at its target point.

Our overarching hypothesis again is that poising the dynamics of neuronal ensembles at such dynamically critical states endows the system with two essential properties: the ability to flexibly deploy and retract different modes of behavior and change function or connectivity, as well as integrating neural ensembles and even entire brain regions into coherent wholes in input-dependent or context-dependent manner. We specifically hypothesize that a large fraction of what is known as “ongoing brain activity” or “default mode network activity” [88] may be the byproduct of such critical poising; thus spontaneous activity is a delicate and highly improbable dynamical state that has to be actively maintained and protected as it is directly responsible for endowing the system with its flexibility and integrative properties. From this hypothesis, theoretical predictions can be made about how, mechanistically, perturbations to the ongoing activity state would affect integrative brain function.

## 5. Conclusions

A large number of biological systems show dynamics that has been linked to, more or less directly, the system spontaneously poising itself at the boundary of a large number of dynamical transitions. We have reviewed the evolution of these ideas and their firm rootings in experimental evidence. We have derived from this a family of models, the critically coupled map lattices. We have here shown the direct similarity and many connections to a related notion of criticality, that of “edge of chaos” dynamics, connected at its root to cellular automaton Turing universality.

## Figures and Tables

**Figure 1 entropy-24-00591-f001:**
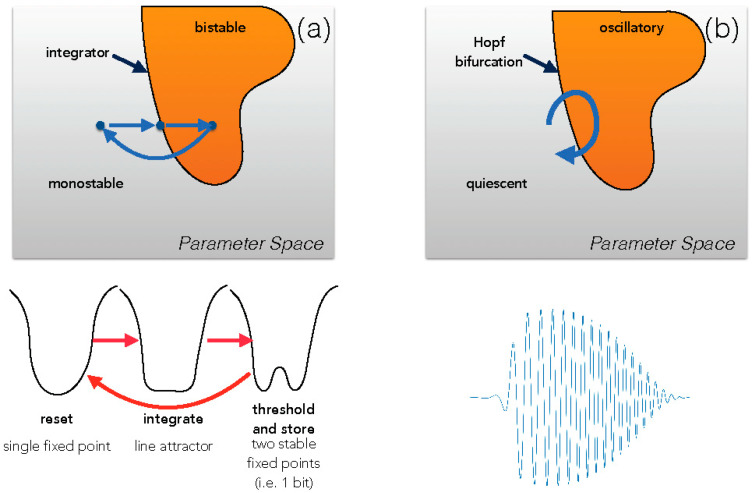
After [3]. Motion through a parameter space, when poised close to a bifurcation, allows rapid reconfiguration of the underlying dynamics. (**a**), a set of transitions from monostable through flat bottom through double-well potential, as in [28]. (**b**), motion near a Hopf bifurcation allows for starting and stopping complex waveforms, as in [25,27].

**Figure 2 entropy-24-00591-f002:**
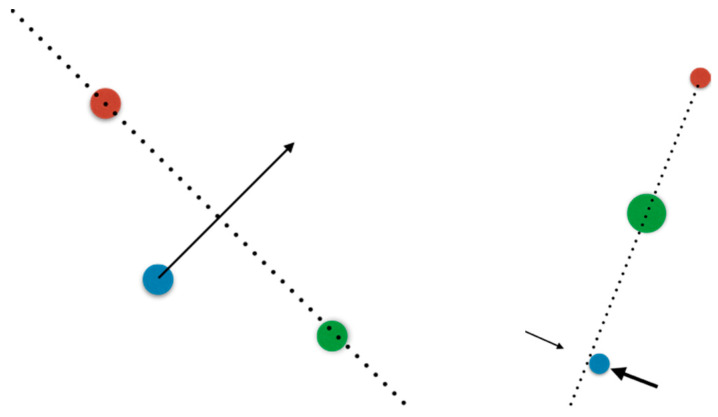
(**Left**): three points on a plane are not typically collinear, so we say collinear configurations have zero probability. However, if the configuration is evolving, *eventually* one point will cross the line defined by the other two; so, it is *certain* that at some rare moments in time the configuration will be collinear. In the case of intersections, this will happen in any situation in which the topology changes. For example, at the instant a falling drop detaches the intersection cannot be transverse. (**Right**): there is a situation far more prevalent in biology, in which there is an active tampering with the configuration: a feedback loop stabilizing the system in the unlikely case. For example, consider a kid playing hide-and-seek; if another kid hides behind a tree, they tries to keep themselves, the seeker and the tree collinear. An archer aiming at a target tries to line up 4 points in 3D: their pupil, the mark, the front of the aim, and the back of the aim (codimension 4). This mechanism can stabilize rare configurations of arbitrarily high codimension. However, the flip side is that the system will not be exactly at the unlikely configuration but forever hovering close to it.

**Figure 3 entropy-24-00591-f003:**
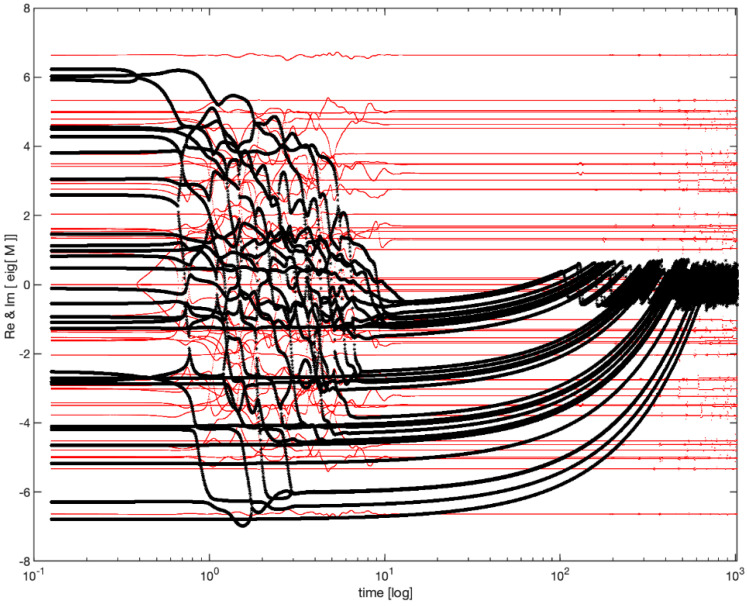
Anti-Hebbian evolution and poising near criticality (following [49]). The temporal evolution of the real (black) and imaginary (red) parts of the eigenvalues of Equation (1). Please note the logarithmic time scale permitting the visualization of distinct epochs. During an epoch of order 1, all unstable eigenvalues (i.e., those with a positive real part) relax to become stable (negative real part). During an epoch of time 1/α , all stable eigenvalues approach zero. Forever thereafter all eigenvalues fluctuate around the critical line, perpetually flirting with instability. Of key importance to our tenets, the timescale in which these eigenvalues fluctuate is neither 1 nor 1/α, but rather their geometric mean $1/\sqrt{\alpha }$.

**Figure 4 entropy-24-00591-f004:**
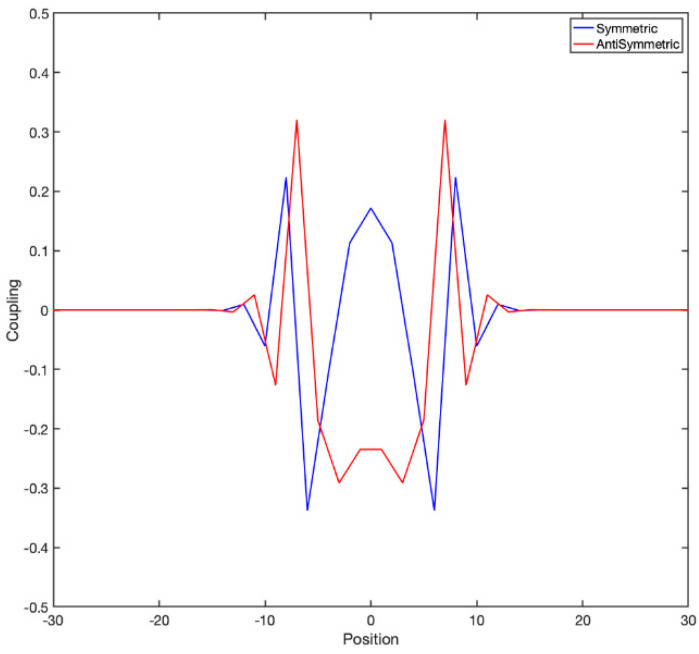
The exponential of the coupling matrix has interesting internal structure. It has both a symmetric and antisymmetric component (as a matrix), corresponding to the hyperbolic cosines and sines; both are spatially symmetric, and each one couples to the odd or the even sublattice. Here is a single row of the matrix exponential of *M* with τ=4. As τ increases the couplings become broader and broader, like in diffusion; however these coupling are such that the determinant of the matrix, instead of asymptoting to zero, stays precisely 1, i.e., conserves phase space volume.

**Figure 5 entropy-24-00591-f005:**
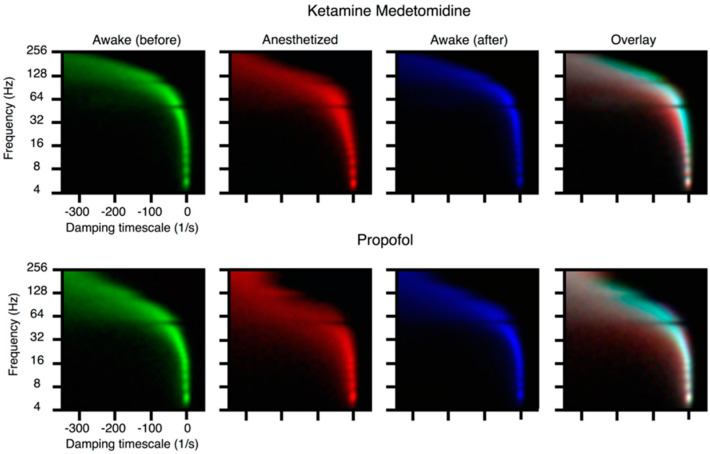
From [67]. Loss of consciousness is associated with stabilization of cortical dynamics. A mechanistic connection between ongoing neural activity and brain function is revealed by dynamic mode analysis of ongoing activity during anesthetic induction. Induction of anesthesia (propofol) in monkeys (see [72] for similar studies in humans) was studied by critical mode analysis [68] of the ECoG recordings of their brain activity, at distinct phases in the induction process. The eigenvalues of the autoregressive process are complex; the real part (horizontal axis) is the growth rate of the mode (negative means stable) while the imaginary part (vertical axis) is the frequency of oscillation associated to the mode. Three two-dimensional histograms of such eigenvalues are colored according to the stage of anesthetic induction: green indicates eigenvalues from a fully conscious subject before induction, red from the fully unconscious subject, and blue the eigenvalues for the conscious subject post-recovery. The rightmost panel shows a merge of these three histograms respecting their colors, where a cyan plume can be seen demonstrating that the activity pre- and post- induction is similar, while the red component shows a motion of the real part of the eigenvalues towards the left (higher stability, thus less responsivity). High-frequency modes in ECoG have been implicated in integrative, cognitive functions [8].

**Figure 6 entropy-24-00591-f006:**
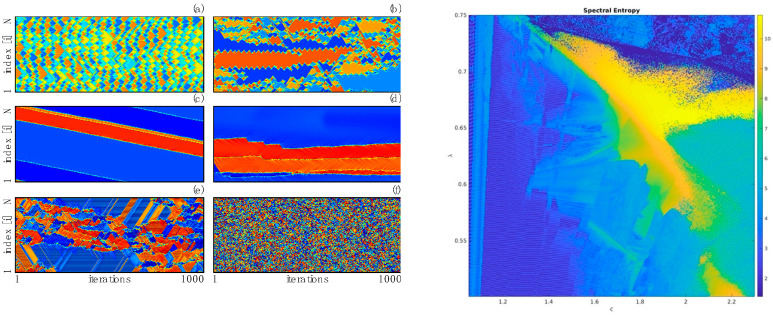
(**Left**), some representative dynamics from this 1D CCML showing in panels (**a**–**f**) a number of different dynamical structures for different parameters values of the model; in each panel time is horizontal and lattice position is vertical. (**Right**), the entropy of spatiotemporal power spectra of the map. The parameter c is the multiplicative parameter in the map $f\left(x\right)=cxe^{-\frac{x^{2}}{2}}$ controlling the transition from order to chaos, while the parameter λ is the “time” during which the spatial coupling M is allowed to act to generate U. In Appendix A, you will find a slice of this phase diagram for λ=0.675 (a horizontal line near the top of the diagram); each frame in the movie is a simulation for a parameter value of c.

**Figure 7 entropy-24-00591-f007:**
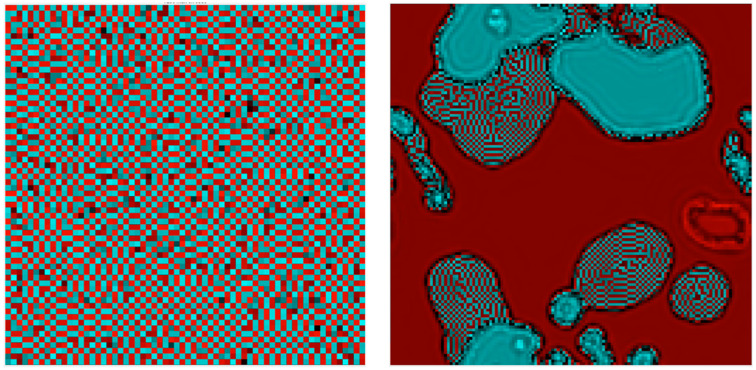
Bipartite 2D model, Gaussian coupling. One asymptotic behavior recurring in this model is shown in the **left** panel (λ=0.5,c=−1.2): a background of frozen disorder (bright red/cyan squares) sees a number of “gliders” (darker areas) move around and re-write the background. Notice that unlike models such as Conway’s Game of Life, the “background” on which these gliders move is information-rich. Please see Appendix A for the time-evolution of this CCML. On the **right**, in a different region of parameter space (λ=0.53,c=−2.3), several distinct background states cyclically lose stability to each other. Please see Appendix A for the time evolution.

**Figure 8 entropy-24-00591-f008:**
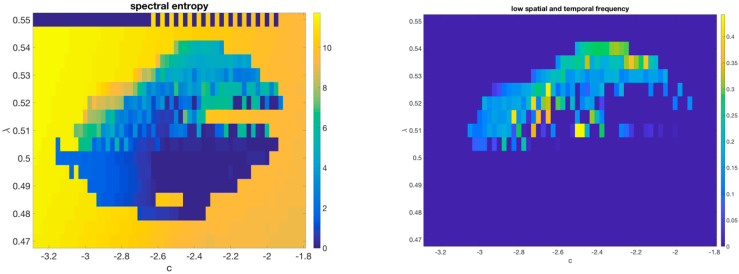
Spectral entropy (**left**) and low-frequency power (**right**) of the 2D model. Please see Appendix A for a slice through the λ=0.525 line displaying the different kinds of behavior available to this CCML.

## Data Availability

Not applicable: all data quoted is available through the cited primary experimental publication.

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
