# Peer review of "Robustness and Flexibility of Neural Function through Dynamical Criticality"

_entropy, 2022, doi:10.3390/e24050591_

Round 1
Reviewer 1 Report
1. The topic is interesting but this article is more like a review paper for the author's former works. There are 74 references but 13 cited papers are written by the author himself. The figures are most directly copied from the ones published by the author's former works. All the examples are from the author's former works.
2. The statement in the "Materials and Methods" and "Results" are fragmented without a primary topic. It is not suitable to call "Methods" and "Results" in a review article. And it's hard to get the core points of the paper.
3. The citations in line 361 and 365 are not correct.
4. For the dynamical criticality, there is a data-driven approach based on bifurcation theory to quantify the criticality or tipping point, i.e. dynamic network biomarker, e.g.
[1] Chen et al. Scientific Reports. 2, 342; DOI:10.1038/srep00342, 2012.
[2] Shi et al. Frontiers in Network Physiology, https://doi.org/10.3389/fnetp.2021.755685, 2022.
Author Response
For both reviewers: due to a cloud storage glitch the version submitted was the immediate prior to my final revision and a number of citations were not properly inserted. I apologize.
I thank Reviewer 1 for their attentive reading. Indeed the manuscript is mostly a review, with some new results. Although this ms was submitted as a review to a journal that does accept reviews, there was no explicit MS Word template for the review format so I worked with what I had. I presume editorial staff will change the headings or direct me as to how to reformat. I did use the M&M section to review the methods themselves as opposed to the results.
As to the number of my own citations, having worked in this subject as a theorist, a data analysis and an experimentalist, for literally 30 years, I do not feel that 17 of my own references in a total of now 88 refs fails to adequately frame this work within a larger setting, particularly since there are major reviews of either the SOC camp or the “edge of chaos” point of view but none from the self-tuned criticality area which I present. This review was long overdue.
Figure 1,2,3,5 are from former work, but 1,2,3 are merely schematics, only 5 shows data and Figure 3 was recomputed from scratch. Figures 4,6,7,8 and the supplementary movies S1-S5 are all original and priorly unpublished. The two-dimensional vpCML models have never been published even as preprints.
I also thank the reviewer for pointing out the interesting references from the Aihara lab, which we now cite. Would like to point out that Shi et al are inferring stabilization of critical modes in pathologies based on an autoregressive fit followed by eigenvalue analysis very similar to our own; I will contact the authors to discuss possible synergies between our approaches.
I believe this answers all the points by the Reviewer in full.
Reviewer 2 Report
Magnasco reviews the concept that neural systems poised near a dynamical bifurcation, at the “edge of chaos”, may confer the systems with both flexibility and robustness. Magnasco focuses on results from his previous work (with colleagues) relating dynamical systems to measured brain dynamics during anesthesia and to simple coupled map lattice models. Generally, the review is clear and well written. I have a few suggestions for improvements.
- In the introductory review of concepts related to criticality in neural systems, I think the author has missed a substantial part of the research community. This is the hypothesis that the brain may operate in a critical state, but with nothing to do with “self-organization” in the sense promoted by Per Bak and colleagues. In other words, the mechanisms of self-organization in sand pile models are unlikely to be relevant in a real brain. Discovering real biological mechanisms of self-organization is part of the experimental challenge for this field, rather than part of the theoretical hypothesis. For example the following papers all operate within this framework of the critical brain hypothesis, without the unlikely SO part of SOC:
- Kinouchi, O. & Copelli, M. Optimal dynamical range of excitable networks at criticality. Nat. Phys. 2, 348–351 (2006).
- Shew, W. L. et al. Adaptation to sensory input tunes visual cortex to criticality. Nat. Phys. 11, 659–663 (2015).
- Suárez, L. E., Richards, B. A., Lajoie, G. & Misic, B. Learning function from structure in neuromorphic networks. Nat. Mach. Intell. 3, 771–786 (2021).
- Finlinson, K., Shew, W. L., Larremore, D. B. & Restrepo, J. G. Optimal control of excitable systems near criticality. Phys. Rev. Res. 2, 1–7 (2020).
- Ma, Z., Turrigiano, G. G., Wessel, R. & Hengen, K. B. Cortical Circuit Dynamics Are Homeostatically Tuned to Criticality In Vivo. Neuron 104, 655-664.e4 (2019).
- Fagerholm, E. D. et al. Cortical Entropy, Mutual Information and Scale-Free Dynamics in Waking Mice. Cereb. Cortex 1–8 (2016) doi:10.1093/cercor/bhw200.
- There are quite a large number of typos and unfinished references and thoughts on pg 8
- what is the 1/? in Fig 3 caption.
Author Response
For both reviewers: due to a cloud storage glitch the version submitted was the immediate prior to my final revision and a number of citations were not properly done. I apologize.
I thank Reviewer 2 for his kind and thoughtful words. I agree that the search for the homeostatic mechanism poising the brain at criticality is fundamental; in this particular sub-thread this started with Camalet et al and Moreau and Sontag back in the early 2000s, with our own work on anti-hebbian networks giving an example of a high-dimensional feedback system homeostating to crititality.
I further thank Reviewer 2 for the added references, some of which fall within the framework discussed in this review but most of which belong in the “balanced network” camp (in the sense of Turrigiano), which I comment upon in the “Relation to other forms of criticality”. For example and to be clear, Ma et al assess criticality through analysis of avalanche-like spike dynamics rather than through crossings of the imaginary axis of eigenvalues, using a method pioneered by my former postdoc J Touboul together with A Destexhe. I now reference all of them.
As discussed above the version submitted did not include my last batch of changes; typos and unfinished references were corrected. Thank you for catching the 1/?; it was a 1/alpha that mistranslated and somehow I never saw it.
I believe this answers all the points by the Reviewer in full.
Round 2
Reviewer 1 Report
No further comment